# Construction of Electrochemical Immunosensor Based on Gold-Nanoparticles/Carbon Nanotubes/Chitosan for Sensitive Determination of T-2 Toxin in Feed and Swine Meat

**DOI:** 10.3390/ijms19123895

**Published:** 2018-12-05

**Authors:** Yanxin Wang, Liyun Zhang, Dapeng Peng, Shuyu Xie, Dongmei Chen, Yuanhu Pan, Yanfei Tao, Zonghui Yuan

**Affiliations:** 1National Reference Laboratory of Veterinary Drug Residues (HZAU), Huazhong Agricultural University, Wuhan 430070, China; yxw@webmail.hzau.edu.cn (Y.W.); zhangliyun@webmail.hzau.edu.cn (L.Z.); pengdapeng@mail.hzau.edu.cn (D.P.); Snxsy1@126.com (S.X.); chendongmei@mail.hzau.edu.cn (D.C.); panyuanhu@mail.hzau.edu.cn (Y.P.); yuan5802@mail.hzau.edu.cn (Z.Y.); 2Key Laboratory of the Detection for Veterinary Drug Residues, Ministry of Agriculture, Wuhan 430070, China

**Keywords:** T-2 toxin, electrochemical immunosensor, gold nanoparticles/single-walled carbon nanotubes/chitosan, feed, swine meat

## Abstract

T-2 toxin (T-2) is one of major concern mycotoxins acknowledged as an unavoidable contaminant in human foods, animal feeds and also agriculture products. Thus, a facile and sensitive method is essential for accurate T-2 toxin detection. In our work, a specific electrochemical immunosensor based on gold nanoparticles/carboxylic group-functionalized single-walled carbon nanotubes/chitosan (AuNPs/cSWNTs/CS) composite was established. The mechanism of the electrochemical immunosensor was an indirect competitive binding to a given amount of anti-T-2 between free T-2 and T-2-bovine serum albumin, which was conjugated on covalently functionalized cSWNTs decorated on the glass carbon electrode. Afterwards, the alkaline phosphatase labeled anti-mouse secondary antibody was bound to the electrode surface by reacting with the primary antibody. Lastly, alkaline phosphatase catalyzed the hydrolysis of the substrate α-naphthyl phosphate, which produced an electrochemical signal. Compared with conventional methods, the established immunosensor was more sensitive and simpler. Under optimal conditions, this method could quantitatively detect T-2 from 0.01 to 100 μg·L^−1^ with a detection limit of 0.13 μg·L^−1^ and favorable recovery 91.42–102.49%. Moreover, the immunosensor was successfully applied to assay T-2 in feed and swine meat, which showed good correlation with the results obtained from liquid chromatography-tandem mass spectrometry (LC-MS/MS).

## 1. Introduction

Species of the genus Fusarium embody the most destructive fungal plant pathogens and are responsible for major yield losses during the cultivation of wheat, maize, barley, and soybeans [1,2]. T-2 toxin is one of major concern mycotoxins, a group of type A trichothecenes secondary metabolite produced by several fungalgenera, including Fusarium fungi, when they infect grains, especially oats and wheat [3,4,5]. Reportedly, it also has been acknowledged as an unavoidable contaminant in human foods, animal feeds and agricultural products such as maize, wheat and oats among many regions and countries [6]. An acute exposure to high-dose trichothecenes triggers various clinical signs for animals such as diarrhea, vomiting, leukocytosis and hemorrhage accordingly [7]. High dose dietary trichothecenes can induce toxicopathological alterations, shock-like syndromes or even death [8,9]. Long-term exposure to trichothecenes leads animals to anorexia, weight loss, nutritional efficiency reduction, neuroendocrine disturbances and immunoregulation [10]. T-2 toxin has attracted much attention worldwide, so in 2001 the Joint Expert Committee on Food Additives conducted a safety assessment for the toxicity of T-2 and pointed out that the provisional maximum tolerable daily intake (PMTDI) of T-2 was 60 ng·Kg^−1^ [11]. Evidence shows that T-2 toxin is the most toxic compound among the trichothecenes and also a most potent cytotoxic and food-borne mycotoxin [8,9]. These factors make it important to develop a sensitive, specific and rapid method for T-2 detection during food safety control.

In the last few years, various available methods for T-2 analytical assays in foods and feeds have been reported, mainly including mature and sensitive liquid chromatography-tandem mass spectrometry (LC-MS/MS) and high-performance liquid chromatography (HPLC), but they typically demand skilled operators, multistep sample pretreatments and expensive instruments, which hinders their practical applications in detecting food products [12,13,14,15]. Additionally, traditional enzyme-linked immunosorbent assays (ELISA) are also widely used for mycotoxin detection for the high specificity from toxin target antigen and its antibody binding [16,17,18]. However, the immunoassay for small molecular mycotoxins in complex matrix samples still exhibits the disadvantages of serious interference of cross reaction and nonspecific reaction, and frequent false positive results [19]. Thus, an assay possessing high sensitivity, selectivity and low costs is urgently needed for the determination of T-2 toxin in agricultural products and foods at trace levels.

The promising NP-based electrochemical bioanalytical methods can seemingly satisfy the mentioned demands owing to their simplicity, high speed, low cost, high throughput detection possibility and sensitivity, particularly in measurements based on the amperometric test [20]. Bioanalytical applications usually require effective signal amplification strategies directly integrated into the bio-recognition process [21]. For this purpose, nanomaterials are increasingly applied in analytical assays because of their small size providing considerable unique properties. Gold (Au) nanoparticles are one of the most common enhancement tools and play a dominant role in the establishment of highly sensitive bioassays because of its good catalytic activity, unique physicochemical properties, long-term stability, easy of synthesis, and excellent biocompatibility [22,23]. Also, ever since the discovery of carbon nanotubes (CNTs), they have been extensively applied in the process of immunosensor fabrication owing to their unique physicochemical and electrical characteristics [24], such as a high surface-to-volume ratio, good electrical conductivity, and electrode kinetics rapidity [25], which contribute to the improvement of sensors’ sensitivity. The chemical functionalization of CNTs can enhance their solubility and biocompatibility through covalent or noncovalent binding, with various chemical groups allowing CNTs to conjugate with numerous biomolecules [26,27]. In electrochemical immunosensing measurements, CNTs can be directly applied to the electrode surface modification. On the other hand, SWNTs could not stably disperse in aqueous solution because nanotubes have a strong attractive interaction [28]. As reported, chitosan when composited with SWNTs can improve their dispersivity [29]. 

Herein, we modified the glassy carbon electrode (GCE) through electrodeposition of gold nanoparticles and deposition of single-walled carbon nanotubes and chitosan composites to realize multiple signal amplification. On the basis of modified electrode, we further developed a novel electrochemical immunosensor with the mechanism of an indirect competitive reaction between free T-2 and T-2-ovalbumin to a fixed amount of T-2 antibody where the T-2-ovalbumin were conjugated with covalently functionalized nanotubes/chitosan/AuNPs (c-SWNTs/CS/AuNPs). Moreover, in previous research, our lab generated highly sensitive and specific antibodies against T-2 [30]. Finally, the electrochemical reaction signal was obtained via efficiently catalytic activity of ALP-anti-antibody (Alkaline phosphate, ALP) towards enzyme substrate anaphthyl phosphate (α-NP). Besides this, research on the electrochemical techniques for T-2 determination is limited. In this paper, we researched the electrochemical behavior of the modified electrode by cyclic voltammetry (CV) and electrochemical impedance spectroscopy (EIS), optimized the conditions of T-2 detection by difference pulse voltammetry (DPV) and lastly determined the spiked samples by this method. The aim of this study is to provide a sensitive and selective analytical method for quantitative detection of T-2 in feed and animal-derived food.

## 2. Results and Discussion

### 2.1. Identification of the Hapten and Antigen

The hapten was identified by ion trap and time-offlight mass spectrometry coupled with an HPLC system (LC/MS-IT-TOF, Shimadzu, Kyoto, Japan). The calculated *m*/*z* for C28H38O12 was M = 566.58; for [M − H]^−^ the *m*/*z* was 565.2407 which showed that almost all the T-2 toxin was converted to T-2HS (Appendix A). The synthetic antigen was identified with an 8453 UV-Visible spectrophotometer. The results showed that the ultraviolet absorbance spectra of T-2HS-OVA (λmax, 278 nm) was different from that of OVA (λmax, 279 nm), which revealed that the antigen was successfully prepared (Appendix A). 

### 2.2. SEM and AFM Characterization

In the research, the morphological features of modified electrodes were characterized with the scanning electron microscopy (SEM; Figure 1A,B) and atomic force microscopy (AFM; Figure 1C,D). As shown in Figure 1A, the surface of electrodes was dotted with AuNPs uniformly. When modified with cSWNTs/CS, it seemingly shaped a three-dimensional network structure on the GCE surface (Figure 1B). Those characteristics provided a film surface area with great biocompatiblity [31]. The analytical results of AuNPs/cSWNTs/CS and T-2-OVA-cSWNTs/CS/AuNPs by AFM colored graphic were presented in Figure 1C,D, where the location of all kinds of elements were marked by corresponding pixels. As the AFM image of Figure 1D showed, after the immobilization of carboxylated SWNTs/CS/AuNPs onto the electrode surface by diimide activation, several changes happened in the superficial morphology. When comparing the AFM image of T-2-OVA-cSWNTs/CS/AuNPs and that of AuNPs/cSWNTs/CS, the characterized result of AFM showed a tidy and closely image in Figure 1D, implying that T-2-OVA was successfully immobilized onto the cSWNTs/CS/AuNPs film.

### 2.3. Characterization of the Immunosensor

In this research, both cyclic voltammetry (CV) and electrochemical impedance spectroscopy (EIS) methods were adopted to test the interface features of the stepwise modified immunosensor. Figure 2A showed the CV results of different fabricated electrodes performed in 5 mM [Fe(CN)6]^3−/4−^ containing 1 M KCl at 50 mV/s scan rate. When the AuNPs were deposited onto the GCE surface, the redox peak current went up dramatically (curve b), indicating that the AuNPs can accelerate electron transfer speed. After the cSWNTs were dropped on the electrode surface, the redox peak current continuously grew (curve c) because the cSWNTs could promote electron transfer and supplied a larger surface to enrich the loading amount of T-2-OVA. In sequence, when T-2-OVA was dropped onto the activated electrode surface, apparently the peak current decreased owing to the steric hindrance and an obstacle of insulated T-2-OVA for the electron transfer, explaining that T-2-OVA was successfully immobilized on the SWNTs/CS/GCE film surface (curve d). After the immunosensor was incubated with anti-T-2, a further decrease in peak current was produced for the increasing of electron transfer resistance (curve e), implying that we successfully obtained the fabricated electrochemical biosensor for specific recognition and binding with anti-T-2. Lastly when we dropped the diluted ALP-anti-antibody onto the surface of the established electrode, accordingly the redox peak current decreased, further implying that the ALP-anti-antibody was immobilized to the immunosensor successfully. 

In addition, to further research the properties of the desirous immunosensor, we also utilized the procedure of EIS to produce the Nyquist plots composed with a semicircle portion at higher frequencies level corresponding to the electrontransfer-limited process and a linear part at the lower frequency range corresponding to the diffusion-limited process to characterize the whole process of preparing modified electrodes [32,33]. Figure 2B displayed the Nyquist curves of 5 mM [Fe(CN)6]^3−/4−^ containing 1 M KCl at stepwise established electrodes. Afterwards, the bare GCE electrode displayed a quite small semicircle domain indicating a quite fast electron-transfer process (curve a), which was characteristic of a mass diffusion limiting step in the electron-transfer process. Because of the conductivity of AuNPs/SWNTs/CS, an apparent decrease of semicircle diameter was achieved after the step of electrochemical deposition (curve b and c). The built film contributed to promoting the electron transfer rate of the [Fe(CN)6]^3−^/[Fe(CN)6]^4−^ couple and obtaining a larger interface area, herein improving the loading amount of T-2-OVA [32,33,34,35]. However, a stepwise growth in semicircle diameters corresponding with the increased resistances happened after the sequential fabricating of T-2-OVA, Anti-T-2 and ALP-anti-anti-body (curves d, e and f separately). The reason for that was the protein layer acting as the inert electron, while the mass-transfer blocking layer significantly deterred the diffusion of ferricyanide towards the electrode surface [24,28]. 

### 2.4. Sensor Optimization

In order to achieve the best analytical property for T-2, we optimized several experimental key parameters. As displayed in Figure 3A, we recorded the peak current response, which increased to a stable value at the concentration of immobilized T-2-OVA of 10 μg·mL^−1^. Therefore, we selected 10 μg·mL^−1^ as the optimal concentration. When the amounts of T-2-OVA domains on the electrode surface was fixed, the concentration of anti-T-2 in the reacting solution was another crucial influencing factor because the competitive immunoreaction exists between the fixed antigen domains and the free T-2 in solution to bind to anti-T-2. For getting the optimal concentration of anti-T-2, it was necessary to incubate the fabricated T-2-OVA-AuNPs/SWNTs/CS/GCE in different concentrations of anti-T-2 solutions. As displayed in Figure 3B, the signal response rose highest at 5 μg·mL^−1^. Therefore, we chose 5 μg·mL^−1^ as the performed concentration in this assay. 

In addition, the dilution ratio of ALP-anti-antibody greatly affected the enzymatic reaction rate. In Figure 3C, the DPV response gradually increased with the reducing dilution ratio of ALP-anti-antibody from 1:3200 to 1:200 and gradually trended to level off at 1:800. Thus 1:800 was used in the experiment as the optimal ALP-anti-antibody ratio. At last, the concentration of substrate α-NP would be another important parameter influencing the enzyme-catalyzed reaction when the dilution ratio of ALP-anti-antibody was regular [36]. In Figure 3D, the signal enhanced continuously with the concentration of α-NP in DEA solution increasing. As the concentration of α-NP was over 0.75 μg·mL^−1^, the further increase of response signal was not obvious, so 0.75 μg·mL^−1^ was adopted as the final work concentration.

### 2.5. Linear Range and LOD

To investigate the performance of this electrochemical immunoassay, different concentrations of T-2 accurately prepared with PBS (pH 7.4) were tested, and the resulting DPV responses were shown in Figure 4A. The peak current values exhibited a linear decrease with the logarithm of T-2 concentrations from 0.01 to 100 μg·L^−1^, the correlation coefficient was 0.9986 (Figure 4B), and the error bars represented standard deviations of three independent measurements.

The LOD of this sensor was calculated using a linear regression curve based on the method reported in literature [33,37]: LOD = 3 × Sa/b. For this method, Sa is the standard deviation of the intercept and b is the slope of the calibration curve. Using the linear regression y = −54.838x + 137.57 presented in Figure 4B, the calculated LOD values for T-2 was 0.14 μg·L^−1^ (linear range: 0.01 to 100 μg·L^−1^, R^2^ = 0.9986). 

As compared in Table 1, the LOD value was lower and the obtained linear range of toxin concentration was wider than those from other reported tests, such as ELISA and LC-MS/MS [30,38,39], further highlighting the merits of the suggested sensor.

### 2.6. Specificity, Reproducibility and Stability

The specificity of the immunosensors was tested by examining 10 μg·L^−1^ T-2 toxin and other three mycotoxins including Deoxynivalenol (DON), Nivalenol (NIV), Neosolaniol (NEO) respectively. As displayed in Appendix A, the DPV signal values of the three different mycotoxins were similar to that of the blank (without T-2). On the contrary, the DPV signal of T-2 showed a sharp reduction and the error bars represented the standard deviation for three measurements, which revealed that the assembled sensor could effectively analyze the target toxin owing to the specific recognition mechanism between antigen and antibody.

To evaluate the reproducibility of this developed immunoassay, carried out five reduplicative tests of T-2 at 0.1 μg·L^−1^ and 100 μg·L^−1^ with the same batch of immunosensors. The coefficients of variation were 5.73% and 8.26%, separately implying that the designed immunosensor using AuNPs/SWNTs/CS as a signal amplification strategy confirmed good reproducibility.

The stability was evaluated by recording DPV current signals of T-2 toxin after storing the electrode at 4 °C for up to two weeks. During the first week, the current response showed a slight change; however, a somewhat larger decrease was observed during the next week (Appendix A). Overall, the catalytic current decreased to about 70% of its initial value and an RSD was less than 7.02% (*n* = 3), indicating good stability over two weeks. This result expressing optimistic stability of the immunosensor is due to the fact that the platform was relatively stable and both the attachment of antibodies onto the electrode surface and the affinity bonding between antibody and antigen were relatively firm [28]. On the other hand, the gradual decline in the current value may be due to the slow denature of biomolecules [40].

### 2.7. Sensor Performance in Extracts from Spiked Maize and Swine Meat Samples

To validate the applicability of the presented method for specifically and sensitively detecting T-2, the spiked feed and swine meat samples at concentrations of 10, 50, 200 μg/kg were assessed with the suggested method. The recovery of spiked toxins in feed and swine meat samples were from 91.59% to 102.49%, 91.42% to 100.80% and their RSD results were lower than 10% (Table 2). These results indicated very high acceptability of the designed sensor for detecting mycotoxin in food matrices.

## 3. Experimental Section

### 3.1. Reagents

T-2, ovalbumin (OVA) were purchased from Beijing Huaan Magnech Biotechnology Co., Ltd. (Beijing, China). Gold chloride (HAuCl4, Chitosan, ALP-anti-anti-body, *N*-hydroxysuccinimide (NHS), α-naphthyl phosphate (α-NP) and 1-ethyl-3-(3-dimethyllaminopripyl) carbodiimide hydrochloride (EDC) were from Sigma-Aldrich (New York, NY, USA). Carboxylic group-functionalized SWNTs (<5 nm diameter) were obtained from Shenzhen Nanotech Port Co., Ltd. (Shenzhen, China). Other reagents were of analytical grade and all aqueous solutions were prepared using Millipore-Q water (18 MW). 0.01 M phosphate-buffered saline (PBS), as washing buffer, contained 2.7 mM KCl, 14 mM KH2PO4, 87 mM Na2HPO4 and 136.7 mM NaCl (pH 7.4). Diethanolamine (DEA) buffer contained 0.1 M DEA, 1 M MgCl2 and 100 mM KCl (pH 9.6). All other chemicals, such as Succine anhydride and *N*,*N*-dimethylformamide (DMF), were of analytical grade and were purchased from Sinopharm Chemical Reagent Co., Ltd. (Shanghai, China).

### 3.2. Apparatus

All the electrochemical experiments including differential pulse voltammetry (DPV), cyclic voltammetry (CV) and electrochemical impedance spectroscopy (EIS) were performed with Autolab PGSTAT128N (Vanton, Switzerland). All the measurements were carried out with a conventional three-electrode system that composed of an Ag/AgCl saturated with KCl as reference electrode, a platinum wire as auxiliary electrode and a 3-mm-diameter GCE modified with the AuNPs/SWNTs/CS as working electrode at room temperature. The scanning electron microscopic (SEM) images and atomic force microscope (AFM) images were photographed with the SU8010 scanning electron microscope (Hitachi, Japan) and Multimode 8 (Cambridge, MA, USA).

### 3.3. Electrochemical Deposition of AuNPs onto the GCE

Firstly, the glass carbon electrode (GCE, 3 mm diameters) was polished with 0.3 and 0.05 µm alumina powder respectively and then rinsed completely with absolute alcohol and distilled water in ultrasonic bath. After being dried at room temperature, the GCE was immersed in 1 g/L auric chloride acid aqueous solution and afterwards AuNPs were deposited by chronoamperometric method with a voltage of −0.2 V and a time of 200 s.

### 3.4. Preparation of GCE Modified with AuNPs/SWNTs/CS

5 mg carboxylic group-functionalized SWNTs fine powders and 10 mg CS were dispersed into 10 mL *N*-*N*-dimethylformamide and sonicated to black uniform mixture. We stored the SWNTs/CS dispersion liquid at 4 °C. 10 µL of the mixture was drop-casted onto the GCE surface modified with nano-gold and dried at room temperature overnight.

### 3.5. Synthesis of T-2HS Hapten and T-2HS-OVA Antigen Hemisuccinate

The T-2-hemisuccinate (T-2HS) hapten was synthesized from HS to create a reactive group by modifying the procedures from a previous study [41,42]. Briefly, 5 mg of T-2 was dissolved in 2 mL pyridine and then 80 mg of succinic anhydride was added. The mixture was in a oil bath at 60 °C for 48 h and was dried with a flow of nitrogen. The residue was redissolved in 2 mL of chloroform and washed with distilled water four times. The chloroform extract was subsequently dried with a flow of nitrogen to obtain the T-2HS conjugate.

The hapten was transferred into 500 µL dimethylformamide (DMF) containing 5 mg DCC and 3 mg NHS and the mixture was stirred at RT overnight. Then the reaction solution was added drop-wise into the 5 mL of 0.1 mol·L phosphate buffer (PBS, pH 7.4) containing 20 mg OVA in an ice-bath and under continuous stirring for 24 h. Lastly, put the solution into a dialysis bag and dialyzed in PBS buffer solution for 3 days. Afterwards, we centrifuged the mixture for 10 min at 5000 rpm and reserved the supernatant at −20 °C prior to use.

### 3.6. Preparation of Electrochemical Immunosensor

We prepared the electrochemical immunosensor using the following steps: (i) As 2.3 mentioned; (ii) As 2.4 mentioned; (iii) After PBS clearing, we immersed the Au/SWNTs/CS/GCE into 50 µL freshly prepared EDC and NHS solution (5 mM:2 mM) and incubated it in a 37 °C incubator for 1 h to activate the carboxylic group of SWNTs; (iv) After the activation, we thoroughly washed Au/SWNTs/CS/GCE with PBS. Following this, we immediately dropped 10 µL of 15 μg·mL^−1^ T-2-OVA onto the electrode surface and incubated it for 1 h at 37 °C; (v) Then, we rinsed the built electrode with PBS and next treated it with 10 µL 2% BSA solution for 1 h at room temperature to block the inactive sites; (vi) Finally, we thoroughly rinsed the electrode with PBS again. The electrochemical immunosensor based on SWNTs/CS/AuNPs/GCE was successfully prepared and stored at 4 °C prior to use. The fabrication process for this electrochemical immunosensor was displayed in Scheme 1.

### 3.7. Electrochemical Measurements

In order to perform the competition assay, firstly blended 5 µL of the diluted anti-T-2 (5 μg·mL^−1^) with 5 µL of T-2 standard solution, with concentrations from 0 to 500 ng·mL^−1^ in PBS. We added the miscible liquids onto the modified T-2-OVA electrochemical immunosensor surface and then incubated it at 37 °C for 1.5 h. In the incubation process, a competing reaction happened between the immobilized T-2-OVA and free T-2 toward a rationed anti-T-2 in the mixture. After being rinsed with PBS, we incubated the prepared immunosensor with 10 μL of diluted ALP-anti-antibody (1:800, *v*/*v*) for 1.5 h at 37 °C. We washed the resulting electrode again with PBS and finally immersed it into the freshly prepared DEA solution containing 0.75 mg·mL^−1^ α-NP. We performed the differential pulse voltammetry (DPV) measurements at the setting of 0.05 s modulation time, potential scan from −0.2 to 0.8 V, pulse amplitude of 70 mV, a pulse period of 0.2 s and interval time of 0.017 s at room temperature.

### 3.8. Sample Preparation

We obtained swine feed from the Breeding Swine Testing Centre (Huazhong Agriculture University, Wuhan, China). Firstly, we sifted them through a 40-mesh sieve, weighed a total 2 g of every finely ground samples into 50 mL tubes and added 10 mL of methanol-water (7:3, *v*/*v*). After a 5 min vortex, we then filtered the samples through Whatman No.1 filter paper. After that, we diluted 1 mL of the filtrate into 4 mL of PBS (0.01 mol·L^−1^, pH 7.4) and filtered it again for analysis.

We purchased minced and finely homogenized swine muscle samples from a local market. We weighed 2 g of every sample into 50 mL tubes, and added 10 mL of methanol-water (7:3, *v*/*v*), and then centrifuged at 4000 rpm for 10 min after a 5 min vortex. In sequence, we removed 1 mL of the supernatant, then diluted into 3 mL of PBS and lastly added 1 mL hexylhydride into the lower layer after vortexing and standing for the electrochemical analysis.

## 4. Conclusions

In conclusion, the electrochemical immunosensor based on AuNPs/SWNTs/CS integrated with an enzymatic signal readout mechanism displayed excellent properties for the detection of T-2. The double signal amplification strategy on the AuNPs/SWNTs/CS prominently improved the sensitivity of T-2 determination. Simultaneously, the ALP possessing excellent catalytic activity towards α-NP enhanced the sensitivity of this assembled immunosensor. The developed method showed optimistic establishment reproducibility and a relative excellent specificity and sensitivity for T-2 detection as well. It also additionally exhibited a wide liner range from 0.01 to 100 ng·mL^−1^ with the limit of detection of 0.14 μg·L^−1^. In summary, the suggested strategy provides an immobilized and sensitized recognition platform for T-2, which could be a promising application for T-2 detection in food samples and hopefully could further furnish a versatile and forceful tool to guarantee food safety in the future.

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
