# Peer review of "Construction of Electrochemical Immunosensor Based on Gold-Nanoparticles/Carbon Nanotubes/Chitosan for Sensitive Determination of T-2 Toxin in Feed and Swine Meat"

_ijms, 2018, doi:10.3390/ijms19123895_

Round 1

Reviewer 1 Report

This is an interesting use of gold nanoparticles and chitosan to detect T-2 toxin.  It and other related research will serve as a foundation for the scientific community to develop better methods to detect toxins and other analytes based on nanotechnology.  The research will be of interest to nanotechnology researchers, analytical chemists, food scientists, and mycotoxin researchers.

Author Response

Response to Reviewer 1 Comments

This is an interesting use of gold nanoparticles and chitosan to detect T-2 toxin.  It and other related research will serve as a foundation for the scientific community to develop better methods to detect toxins and other analytes based on nanotechnology. The research will be of interest to nanotechnology researchers, analytical chemists, food scientists, and mycotoxin researchers.

Response: Thanks for your praise so much that we earnestly hope the research will be of interest to nanotechnology researchers, analytical chemists, food scientists, and mycotoxin researchers. Thanks for your high comments again.

Besides, we also checked English language and style and made some minor spell correction in red words.

Reviewer 2 Report

The manuscript described the development of the electrochemical immunosensor based on AuNPs/SWNTs/CS integrated with enzymatic signal readout mechanism displayed excellent properties for detection of T-2toxin (T-2). This detective system is more sensitive and simpler than the existing methods. T-2 is one of major concern mycotoxins. Thus, the authors’ novel system is expected to be a promising approach to detect T-2. Therefore, the manuscript is not too excellent to be published. In other words, the manuscript is so excellent that it should be published.

Comments

(1) Can electrochemical immunosensors described here be prepared at low cost?

(2) Is each parts of electrochemical immunosensors based on AuNPs/SWNTs/CS stable to restore after preparation?

(3) Immunosensor is likely to show lower sensitivity, for example, concerning to limit of detection, than CL-ELISA and HPLC-MS/MS? Can this be improved?

That is all.

Author Response

Response to Reviewer 2 Comments

The manuscript described the development of the electrochemical immunosensor based on AuNPs/SWNTs/CS integrated with enzymatic signal readout mechanism displayed excellent properties for detection of T-2toxin (T-2). This detective system is more sensitive and simpler than the existing methods. T-2 is one of major concern mycotoxins. Thus, the authors’ novel system is expected to be a promising approach to detect T-2. Therefore, the manuscript is not too excellent to be published. In other words, the manuscript is so excellent that it should be published.

Response: Thanks for giving a high appraisal to this research so much that we sincerely hope the mentioned method could be a promising approach to detect T-2. Thanks for your high comments again.

Point 1: Can electrochemical immunosensors described here be prepared at low cost?

Response 1: After the calculation of the single preparation of the electrochemical immunosensor to detect T-2, we believe that the proposed method is at a lower cost than the HPLC-MS/MS method.

Point 2: Is each parts of electrochemical immunosensors based on AuNPs/SWNTs/CS stable to restore after preparation?

Response 2: As illustrated in the third paragraph of 2.6, the stability mainly depends on the attachment of antibodies onto the electrode surface, the affinity bonding between antibody and antigen and the slow denature of biomolecules, so each part shares similar stability to restore after short time preparation. Additionally, the nano-material shows relatively good stability after research.

Point 2:  Immunosensor is likely to show lower sensitivity, for example, concerning to limit of detection, than CL-ELISA and HPLC-MS/MS? Can this be improved?

Response 2: As shown in the table 1, the limits of detection in the listed methods used different measurement units. So after the unit conversion, we drew the conclusion that the LOD value was lower and the obtained linear range of toxin concentration was wider than those from other reported tests, such as ELISA and LC-MS/MS, further highlighting the merits of the suggested sensor. And the aim of this study is to provide a sensitive and selective analytical method for quantitative detection of T-2 using the signal amplification strategy of AuNPs/SWNTs/CS integrated with enzymatic signal readout mechanism, finally we seemly made it.

Besides, we also checked English language and style and made some minor spell correction in red words.
